# CA9, CYFIP2 and LGALS3BP—A Novel Biomarker Panel to Aid Prognostication in Glioma

**DOI:** 10.3390/cancers16051069

**Published:** 2024-03-06

**Authors:** Amanda L. Hudson, Angela Cho, Emily K. Colvin, Sarah A. Hayes, Helen R. Wheeler, Viive M. Howell

**Affiliations:** 1Bill Walsh Translational Cancer Research Laboratory, Kolling Institute, Royal North Shore Hospital, Northern Sydney Local Health District, St. Leonards, NSW 2065, Australiaemily.colvin@sydney.edu.au (E.K.C.); sarah@gicancer.org.au (S.A.H.); viive.howell@sydney.edu.au (V.M.H.); 2School of Medical Sciences, Faculty of Medicine and Health, University of Sydney, Sydney, NSW 2006, Australia; 3The Brain Cancer Group, North Shore Private Hospital, St. Leonards, NSW 2065, Australia; helen.wheeler@health.nsw.gov.au; 4Department of Medical Oncology, Royal North Shore Hospital, Northern Sydney Local Health District, St. Leonards, NSW 2065, Australia

**Keywords:** glioma, biomarkers, prognosis, CA9, CYFIP2, LGALS3BP

## Abstract

**Simple Summary:**

Glioma is a deadly disease with few treatment options. Early detection of recurrence and progression will help in the clinical management of glioma patients. Biomarkers or molecules that can accurately predict clinical outcomes are becoming extremely important. This study aimed to identify biomarkers of progression and overall survival in glioma. A three-protein panel that could accurately distinguish between long and short survival was identified. Incorporating such biomarkers into clinical practice could significantly aid clinical management.

**Abstract:**

Brain cancer is a devastating and life-changing disease. Biomarkers are becoming increasingly important in addressing clinical issues, including in monitoring tumour progression and assessing survival and treatment response. The goal of this study was to identify prognostic biomarkers associated with glioma progression. Discovery proteomic analysis was performed on a small cohort of astrocytomas that were diagnosed as low-grade and recurred at a higher grade. Six proteins were chosen to be validated further in a larger cohort. Three proteins, CA9, CYFIP2, and LGALS3BP, were found to be associated with glioma progression and, in univariate analysis, could be used as prognostic markers. However, according to the results of multivariate analysis, these did not remain significant. These three proteins were then combined into a three-protein panel. This panel had a specificity and sensitivity of 0.7459 for distinguishing between long and short survival. In silico data confirmed the prognostic significance of this panel.

## 1. Introduction

Brain cancer is a devastating and life-changing disease. While its incidence is low, brain tumours cause the highest economic burden of all cancers, not only for the afflicted individual but also for Australian households [1]. The number of patients surviving five years beyond their diagnosis is unacceptably low, with median survivals of just 5, 3.5, or 1.2 years for grade II, III, or IV astrocytomas, respectively [2]. Patient management primarily consists of maximal surgical resection followed by radiotherapy and temozolomide (TMZ) chemotherapy. Unfortunately, even with treatment, glioma remains an incurable disease wherein progression and recurrence are inevitable [3,4].

Monitoring tumour progression and recurrence and assessment of treatment efficacy are performed through routine imaging approaches such as magnetic resonance imaging or computed tomography scans. However, interpretation of these images is often difficult, as radiological findings which show an increase in tumour size or contrast enhancement may suggest tumour recurrence or may reflect post-treatment-related effects [5,6]. Accurate identification of early progression versus treatment-related changes detected through imaging, particularly in the months following radiation treatment and concurrent chemotherapy, is an important distinction to determine the need for continuing therapy. These clinical challenges highlight the urgent need for advancements in this field. Biomarkers are becoming increasingly important in this regard and are being used as tools to aid clinical practice, aiding in assessing survival and treatment response and, importantly, potentially distinguishing between tumour progression and treatment-related effects. Biomarkers also have the ability to help unravel the biological mechanisms driving these clinical challenges; however, no reliable biomarkers currently exist [7].

The goal of this study was to identify prognostic protein biomarkers in glioma. Discovery proteomic analysis was performed on a small cohort of cohort of six matched pairs of astrocytomas that were diagnosed as low-grade and recurred at a higher grade. Literature searches of the differentially expressed proteins were then performed to narrow down the potential biomarker list to those with published associations with cancers. Five proteins, CYFIP2, F13A1, FABP7, LGALS3BP, and PRKARIA, found to be differentially expressed were then chosen for validation. In addition, CA9, a protein previously reported to be involved in resistance and prognosis in glioma [8,9,10,11,12,13], was also validated in this cohort. Of these six proteins, only three were found to be significantly associated with tumour progression or increasing grade; thus, only these three were then assessed as prognostic markers. When combined into a panel, these three proteins were found to be significantly associated with tumour prognosis according to multivariate analysis.

## 2. Materials and Methods

Ethics: This study was carried out in accordance with the recommendations of the Northern Sydney Local Health District Human Ethics Committee. The protocol was approved by the Northern Sydney Local Health District Human Research Ethics Committee under protocol LNR/17/HAWKE/363. Written informed consent was obtained from all subjects in accordance with the Declaration of Helsinki.

Patient cohorts: The discovery and validation cohorts consisted of patients who presented to Royal North Shore Hospital between 2003 and 2018. The discovery cohort consisted of 6 matched pairs of IDH-mutated astrocytomas that were diagnosed as low-grade and recurred at a higher grade. Clinical characteristics are presented in Appendix A. The validation cohort consisted of 82 fresh-frozen tumours from patients diagnosed with glioma (patient characteristics are summarised in Table 1). To be included, fresh-frozen and formalin-fixed paraffin-embedded (FFPE) tumour tissue and clinical outcome data needed to be available. Initial pathology reports ranged from multiple WHO CNS classification eras. Where possible, available histopathological (e.g., related to microvascular proliferation, necrosis, and mitoses) and molecular data (e.g., concerning *TERTp* mutation, *EGFR* amplification, and/or +7/−10) were used to reclassify all samples based on the current WHO 2021 criteria [14]. CDKN2A homozygous deletion could not be used for classification; however, evidence suggests this correlates with morphological high-grade features (e.g., microvascular proliferation or necrosis) and thus is unlikely to change the classification of the majority of these tumours [15]. IDH status was determined through immunohistochemical screening for p.R132H mutations conducted by Pathology North, RNS Hospital (Clone H09 monoclonal antibody; Dianova, Hamburg, Germany), and IDH pyrosequencing using the Pyromark Q24 pyrosequencing platform (Qiagen) conducted by the Department of Neuropathology (Royal Price Alfred Hospital, Sydney, Australia).

Mass spectrometry (MS): Protein was extracted from fresh-frozen sections via homogenisation in 100 mM TEAB/1% sodium deoxycholate and then concentrated and purified via filter-aided sample preparation using concentrator columns. Proteins were then trypically digested (1 μg trypsin: 50 μg protein), cleaned using C18 tips, and lyophilised. Samples were reconstituted in 2% acetonitrile/0.1% formic acid prior to injection for MS. MS analysis was performed on a collaborative fee-for-service basis at the Australian Proteome Analysis Facility (Macquarie University, North Ryde, Australia) using a TripleToF 6600 + mass spectrometer (SCIEX, Framingham, MA, USA) with an Eksigent nanoLC 400 system with cHiPLC (SCIEX) upfront. A spectral library was first generated using pooled tumour samples and conventional liquid chromatography–tandem mass spectrometry (LC-MS-MS) in Information-Dependent Acquisition (IDA) mode. Sequential Windowed data independent Acquisition of the Total High-resolution Mass Spectra (SWATH-MS) analysis was then performed. A set of 100 overlapping SWATH windows was constructed to cover the precursor mass (MS) range of 400–1250 *m*/*z*. SWATH MS/MS spectra were collected from 350 to 1500 *m*/*z* (accumulation time: 50 ms). MS/MS spectra were accumulated across the *m*/*z* range of 350–1500 (accumulation time: 30 ms) with rolling collision energy. The collision energy for each window was determined based on the appropriate collision energy for a doubly charged (+2) ion with the lowest m/z in the window +10% window size. Data from SWATH-MS were matched against the spectral library to identify peptides (75 parts per million mass tolerance, 10 min retention time window, number of peptides per protein: max. 100, number of transitions per peptide: 6, peptide confidence level ≥99%, and FDR ≤ 1%). Normalisation was then performed (regarding total area and intensity), followed by differential expression analysis using a two-sample *t*-test of the log-2-transformed normalised protein peak areas. The fold change (FC) was calculated as the back-transformed ratio of the means of the sample replicates (log-transformed protein peak areas). Cut-offs used to identify differentially expressed proteins were FC > 2 and *p*-values < 0.05.

Immunohistochemistry (IHC) analysis: Immunohistochemistry (IHC) was performed on 4 µm sections of FFPE glioma tissues (grade II–IV). All antibody details and optimised conditions are listed in Appendix A. Antigen retrieval was performed in antigen retrieval solution (pH 9) (Agilent Technologies, Santa Clara, CA, USA) by using a Pascal pressure cooker at 121° for 30 s and 90 °C for 10 s. This was followed by an endogenous peroxidase block using 0.3% H_2_O_2_ for 5 min. A 1 h primary antibody incubation was carried out, followed by Rabbit Envision (Agilent Technologies) for 30 min. ImmPACT NovaRED^®^ (vector Laboratories, Burlingame, CA, USA) was used for detection according to manufacturer’s recommendations. Positive and negative control tissues and isotype controls were included. Representative images are presented in Appendix A. Staining was assessed by two independent observers blinded to patient outcome. Scores were given as the percentage of tumour cells staining positive multiplied by intensity of staining on a scale of 0 to 3. Discrepancies were resolved by consensus between both scorers. Scores for analysis were calculated as the product of the percentage of tumour cells staining positive multiplied by the intensity (scale of 0–3).

In silico databases: Glioma-BioDP [16,17] was used to assess the expression levels of CA9, CYFIP2, and LGALS3BP in high-grade glioma samples. Kaplan–Meier survival curves were generated individually for CA9, CYFIP2, and LGALS3BP and then as a multi-gene panel. The used program incorporates the TCGA-GBM dataset as well as computational analysis to stratify glioblastoma samples based on the four molecular subtypes identified by Verhaak: classical (C), mesenchymal (M), proneural (P), and neural (N)) [18]. Default options for stratification of samples into two groups were used for the TCGA-GBM samples (n = 197): those with gene expression levels lower than the median and those with gene expression levels higher than median. *p*-values for the significance of the difference between the two resulting curves were also calculated. The STRING database [19] was used to identify predicted protein–protein interactions, both direct and indirect, between the 3 proteins of interest, namely, CA9, CYFIP2, and LGLAS3BP.

Statistical analysis: IHC scores were dichotomised into low and high expression for univariate analysis (Prism 8 software, La Jolla, CA, USA; Appendix A). Factors found to be significant through univariate analysis were assessed together with clinicopathological variable 22s to identify independent prognostic factors (Statistical Product and Service Solutions [SPSS] software V29.0.2.0., Chicago, IL, USA). In addition, to assess whether the triple-protein panel could be used as an independent prognostic factor, patients were dichotomised into two groups (triple-negative/single-positive vs. double/triple-positive) based on the expression of the three biomarkers within each tumour. A tumour was categorised as triple-negative, single-positive, double-positive, and triple-positive when none, one, two, or three of these biomarkers were expressed, respectively (based on the cut-off values presented in Appendix A, where no or low expression was classified as ‘negative’ and any/high expression as ‘positive’). Area Under the Receiver Operating Characteristics (AUROC) analysis was conducted using Prism 8 software.

## 3. Results

### 3.1. Candidate Biomarkers Identified

To identify biomarkers of interest, a small discovery cohort was used. This discovery cohort consisted of six matched pairs of IDH-mutated astrocytomas that were diagnosed as low-grade and recurred at a higher grade. Clinical characteristics are presented in Appendix A. Mass spectrometry analysis identified 1806 proteins across all tumour specimens. Of these, only 32 were found to be significantly differentially expressed when comparing diagnostic to recurrent samples, with the majority found to be downregulated (Appendix A). Due to this small number, basic pathway analysis was limited. However, the TCA cycle and synaptic pathways were amongst the top enriched pathways identified. To further narrow the list of potential biomarkers, a literature search was performed to identify published associations with cancers. Five proteins, CYFIP2, F13A1, FABP7, LGALS3BP, and PRKARIA, were chosen for validation in a larger cohort (Appendix A). Additionally, the CA9 protein was also chosen for validation due to its previously reported functions in glioma treatment resistance and prognosis [8,9,10,11,12,13].

### 3.2. Age, Grade, and IDH Mutational Status Were Significantly Associated with Survival in Validation Cohort

The candidate biomarkers were then examined in a larger independent validation cohort. The clinical characteristics for this validation cohort are presented in Table 1. This cohort consisted of 82 patients diagnosed with astrocytoma (grade II–IV), with a median age of diagnosis of 53 years. Most patients were diagnosed at or below 65 years of age (72.0%) and received chemoradiation (63.4%). Approximately half were found to harbour an IDH mutated tumour (37.83%). As expected for a glioma cohort, being younger (≤65 years) or having a tumour lower in grade or containing an IDH mutation were associated with longer median survival according to log-rank analysis (Table 1; *p* < 0.0001).

### 3.3. CA9, CYFIP2, and LGALS3BP Are Significantly Associated with Tumour Progression

Of the six proteins chosen for validation via IHC, three were found to be associated with tumour progression. CA9, CYFIP2, and LGALS3BP levels were found to significantly increase as tumour grade increased (Figure 1; *p* < 0.05). To determine whether these proteins could also be used as prognostic indicators, expression was dichotomised, and univariate analysis was performed. Table 2 shows that each of the three proteins could be used as prognostic indicators, according to univariate analysis (*p* < 0.01).

### 3.4. Three-Protein Panel as an Independent Prognostic Biomarker

Multivariate analysis was then performed to determine whether CA9, CYFIP2, and LGALS3BP could also be used as independent prognostic markers (Table 3). As individual markers, CA9, CYFIP2, and LGALS3BP lost their significance. However, when combined into a three-protein panel, high expression of two or more of these proteins was found to be an independent prognostic marker (Table 4). The accuracy of this three-protein panel as a prognostic index was then measured using area under the receiver operating characteristics curve (AUC). This led to the prediction of a specificity and sensitivity of 0.7459 for distinguishing between long and short survival (Figure 2).

### 3.5. In Silico Data Confirm Prognostic Significance of 3-Protein Panel

As an independent assessment of the prognostic significance of the three-protein panel, an in silico prognostic tool was used. Kaplan–Meier survival analysis using the TCGA gene expression data showed that the combination of *CA9*, *CYFIP2*, and *LGALS3BP* as a multi-gene panel was prognostically significant (Figure 3; *p* < 0.05). As a prognostic tool can also stratify glioma samples according to their molecular subtype, it was gleaned that this multi-gene prognostic panel also retained prognostic power in the proneural subtype (*p* < 0.05) but not in the other subtypes.

## 4. Discussion

Biomarkers that can be used to monitor patients longitudinally for progression are urgently needed in glioma treatment to enable early therapeutic intervention for recurrent disease. This study identified three proteins, CA9, CYFIP2, and LGALS3BP, that were found to be associated with disease progression and significantly shorter overall survival according to univariate analysis. However, when assessed in a multivariate analysis, as individual biomarkers, these proteins did not remain significant. Therefore, CA9, CYFIP2, and LGALS3BP were combined into a three-protein panel. The combination of the high expression of any two of these proteins (any expression level of CA9 and high CYFIP2 and LGALS3BP expression) was identified as an independent prognostic marker via multivariate analysis. This was then independently assessed using an in silico prognostic tool. Together as a panel, CA9, CYFIP2, and LGALS3BP were able to be used as a multi-gene prognostic index, highlighting the utility of these three molecules for predicting prognoses of glioma patients.

CA9 is a hypoxia-inducible protein that is involved in pH regulation in hypoxic cancer cells (reviewed by [20]). It is transcriptionally regulated through HIF-1 and the MAPK and PI3K pathways [21,22,23], well-known contributors to treatment resistance [24,25,26]. In line with this, knocking down or inhibiting CA9 has been shown to increase the susceptibility of glioma cells to temozolomide [8,12,13]. CA9 has previously been found to be associated with poor prognosis in other cancers (breast, lung, cervical, and renal cancer) and in gliomas, supporting these results [9,10,11,27,28,29].

CYFIP2, also known as p53 inducible protein 121, is a pro-apoptotic protein and member of the cytoplasmic FMR1-interacting protein family, but its function in cancer is not well understood. CYFIP2 is associated with regulation of the actin cytoskeleton, known to play a crucial role in cell migration and invasion [30,31] and, in disulfidptosis, to constitute a newly discovered form of programmed cell death [32,33]. Recent publications suggest that gene expression of CYFIP2 is closely related to patient prognosis and the degree of immune cell infiltrate. However, whether CYFIP2 expression is favourable or adverse appears to be dependent on the specific type of cancer [34,35,36]. In lung adenocarcinoma and pancreatic ductal adenocarcinoma, high gene expression was identified as a favourable prognostic factor, whereas in uterine corpus endometrial carcinoma and oesophageal carcinoma, high gene expression was identified as an unfavourable factor [35]. While both low- and high-grade glioma were analysed in the same study, CYFIP2 expression was not found to be associated with prognosis but rather negatively associated with stromal and immune score parameters, different to most other cancer types. In contrast, Chen et. al. identified that gene expression of CYFIP2, when used together with two additional genes, could be used as a prognostic biomarker in glioma [37], similar to the findings in this study. It must be noted that both these studies used gene expression data rather than protein expression data. Nevertheless, further functional analyses are needed to understand the role CYFIP2 plays in glioma.

LGALS3BP is a multifunctional galectin involved in cell–cell and cell–matrix interactions, adhesion, migration, angiogenesis, and immune response (reviewed in [38]). Similar to CYFIP2, whether its expression is favourable or adverse appears to be cancer-dependent, but the majority of research suggests that high expression is associated with unfavourable clinical outcomes [38]. In cancer, the majority of research has focussed on the secreted versions of LGLAS3BP, with LGALS3BP found in plasma, a proposed biomarker for early detection and overall survival in glioma patients [39,40]. Tissue expression has also been investigated and, as in this study, has been shown to be associated with decreased overall survival [41]. However, this was found to be subtype-dependent and only significant in the proneural subtype. This finding is similar to the in silico data from this study, showing that the prognostic significance of LGALS3BP (when combined with CA9 and CYFIP2) was retained when samples were stratified into the proneural molecular subtype. Interestingly, a recent study by Zhu et al. [42] reported that LGALS3BP expression was associated with the mesenchymal subtype, noting reduced expression in proneural tumours when compared to that in a normal brain. Importantly, Zhu et al.’s study did not show Kaplan–Meier survival curves stratified by subtype, which may be why they did not find it as a prognostic indicator in the proneual subtype, as in He et al.’s work [40] and this study. Zhu et al. did, however, show that when combined into a six-galectin gene panel, LGALS3BP was found to have excellent prognostic and predictive value. Functional analysis of the galectins was also investigated, and roles in glioma stemness maintenance and viability were identified [42]. Such studies highlight the potential of this biomarker when combined in a panel as a prognostic index in addition to the potential of using LGALS3BP targeting as a novel therapeutic option.

Given the significance of these molecules in prognosis, they also have the potential to be drug targets for the development of new treatments. However, further functional analyses are needed, particularly in relation to whether there are any interactions between the three biomarkers. Interestingly, to date, there are no known interactions or functions shared by CA9, CYFIP2, and LGALS3BP (Appendix A), indicating further investigation is warranted.

## 5. Conclusions

In summary, this study employed discovery proteomics for patient tissue and validated the use of a three-protein prognostic panel for predicting outcome for glioma patients. The significance of this panel was further confirmed using TCGA data and a multi-gene prognostic tool. Incorporating such biomarkers into clinical practice could significantly aid clinical management.

## Figures and Tables

**Figure 1 cancers-16-01069-f001:**
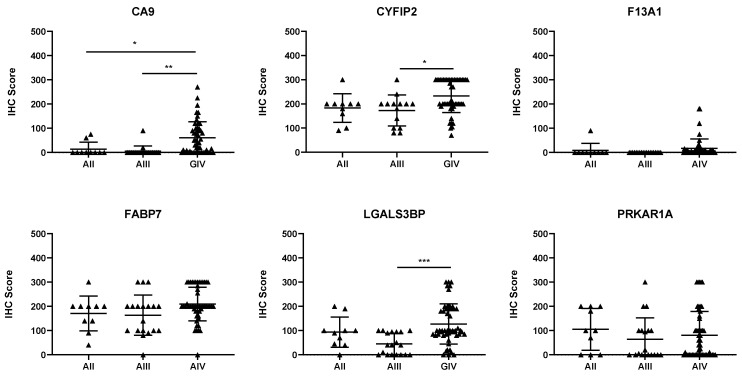
Expression of the candidate biomarkers stratified by glioma grade. Each point represents a single patient. AII, grade II astrocytomas; AIII, grade III astrocytomas; GIV, grade IV astrocytomas. The IHC score is the product of the intensity multiplied by the percentage of positively stained cells. * *p* < 0.05, ** *p* < 0.01, and *** *p* < 0.001.

**Figure 2 cancers-16-01069-f002:**
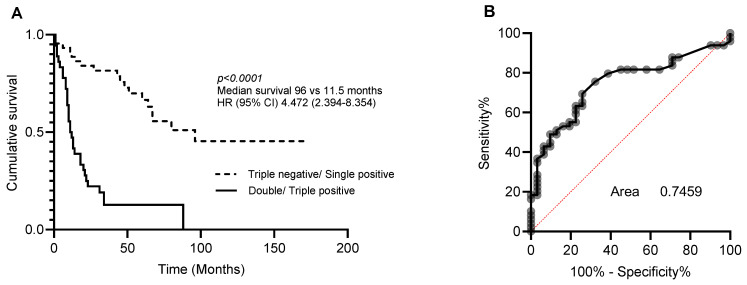
Prognostic significance of 3-protein panel. Patients were dichotomised into triple-negative/single-positive or double-positive/triple-positive for CA9, CYFIP2, and LGALS3BP. (**A**) Kaplan–Meier analysis and (**B**) area under the receiver operating curve (ROC) analysis characteristics curve of patients stratified.

**Figure 3 cancers-16-01069-f003:**
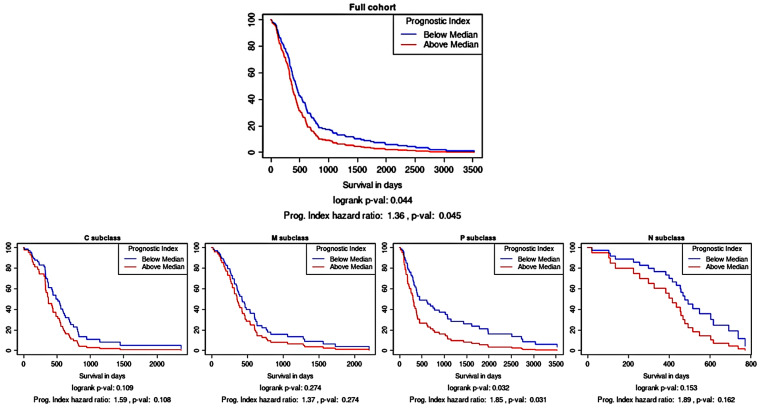
Survival analysis based on the multi-gene (CA9, CYFIP2, and LGALS3BP) prognostic index. Kaplan–Meier survival curves were generated using Glioma-BioD and are presented for the full cohort (197 GBM samples); then, they were stratified according to molecular subtype: classical (C), mesenchymal (M), proneural (P), and neural (N).

**Table 1 cancers-16-01069-t001:** Clinical characteristics of patients in validation cohort. Significant *p*-values (<0.05) have been bolded.

	Number of Patients n = 82 (%)	Median OS (Months)	*p*-Value (Log Rank)
Sex			0.7641
Female	22 (26.8)	23.5	
Male	60 (73.2)	45	
Age (years)			**<0.0001**
Median	53		
Range	20–84		
≤65	59 (72.0)	67	
>65	23 (28.0)	10	
Primary Grade			**<0.0001**
II	10 (12.2)	N/A	
III	20 (24.4)	N/A	
IV	52 (463.4)	13.5	
Treatment			0.097
Nil, chemo or radiation only	8 (9.8)	31.5	
Chemoradiation	52 (63.4)	64	
No treatment info	22 (26.8)		
Outcome ^#^			
Alive	31 (37.8)		
II	8		
III	11		
III	4		
Dead	49 (61.3)		
IDH status			**<0.0001** **IDH MUT v IDH WT**
IDH MUT	39 (47.6)	96	
II	*10*		
III	*20*		
IV	*9*		
IDH WT	43 (52.4)	12.5	

^#^ Censor date, 29 February 2020. N/A, not applicable.

**Table 2 cancers-16-01069-t002:** Prognostic significance of candidate biomarkers determined by univariate analysis. Significant results have been bolded. Protein expression of candidate biomarker was first dichotomised into low and high expression; then, univariate analysis was performed.

	No. of Patients (%)	Median Overall Survival (Months)	HR (95% CI), *p*-Value
CA9 expression			
No expression	41 (50)	80	3.427 (1.91–6.148), **<0.0001**
Expression	41 (50)	13	
CYFIP2 expression			
Low	49 (59.8)	80	2.919 (1.582–5.388), **<0.0001**
High	33 (40.2)	13	
LGALS3BP expression			
Low	43 (53.8)	67	2.073 (1.154–3.724), **0.0071**
High	37 (46.3)	14	

HR, hazard ratio.

**Table 3 cancers-16-01069-t003:** Multivariate analysis of CA9, CYFIP2, and LGALS3BP as individual prognostic indicators.

	Cohort *	
	Hazard Ratio (95% CI)	*p*-Value
Grade		
II	1	0.225
III	2.874 (0.523–15.787)	
IV	2.273 (0.181–28.490)	
IDH (Wildtype)	8.030 (0.866–74.462)	0.067
Age (≥65)	1.020 (0.988–1.053)	0.222
Chemoradiation	0.380 (0.123–1.168)	0.091
CA9 (Expression)	2.301 (0.822–6.439)	0.112
CYFIP2 (High)	1.193 (0.554–2.570)	0.652
LGALS3BP (High)	1.941 (0.866–4.352)	0.107

* 20 patients were excluded due to missing data.

**Table 4 cancers-16-01069-t004:** Multivariate analysis of 3-protein panel as a prognostic indicator. Significant P-values have been bolded (*p* < 0.05).

	Cohort *	
	Hazard Ratio (95% CI)	*p*-Value
Grade		
II	1	0.169
III	3.428 (0.594–19.796)	
IV	2.696 (0.210–34.657)	
IDH (Wildtype)	6.572 (0.697–62.000)	0.100
Age (≥65)	1.017 (0.988–1.046)	0.255
Chemoradiation	0.344 (0.109–1.086)	0.069
≥2 proteins positive	5.113 (1.532–17.064)	**0.008**

* 20 patients were excluded due to missing data.

## Data Availability

The data presented in this study are available in the article and Appendix A.

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
