# Peer review of "CA9, CYFIP2 and LGALS3BP—A Novel Biomarker Panel to Aid Prognostication in Glioma"

_cancers, 2024, doi:10.3390/cancers16051069_

Round 1
Reviewer 1 Report
Comments and Suggestions for Authors
Dear Authors,
I read with the interest manuscript entitled “CA9, CYFIP2 and LGALS3BP - a novel biomarker panel to aid prognostication in glioma.” The study successfully identifies and validates a 3-protein panel (CA9, CYFIP2, and LGALS3BP) as prognostic biomarkers for glioma patients. This can have significant implications for clinical management.
However, I must underline some disadvantages of the paper:
1. The initial discovery cohort is relatively small, consisting of only 6 matched pairs of astrocytomas. While this is a common limitation in early-stage biomarker discovery, it does raise concerns about the robustness and generalizability of the findings.
2. The study mentions patients presented between 2003 and 2018. Given the advancements in diagnostic techniques and treatment modalities over this period, there may be heterogeneity in patient management that could impact the findings. It would be beneficial to explore whether changes in treatment approaches over time influenced the outcomes.
3. The study primarily focuses on glioma patients, and the generalizability of the identified biomarkers to other cancer types is not discussed. It would be essential to explore the specificity of these biomarkers to gliomas.
Thank you very much for your effort to perform the study and prepare the manuscript.
Comments on the Quality of English LanguageThe overall grammar and language usage in the provided text appear to be accurate. However, I noticed a few instances where word choices could be improved for clarity and precision:
In the Abstract, there's a repetition of the phrase "in glioma." Consider rephrasing for variety, such as "associated with glioma progression" or "prognostic markers in glioma."
In the Introduction, the phrase "In-silico data" could be written as "In silico data" for consistency.
In the Results section, under "3.2 Age, grade, and IDH mutational status are significantly associated with survival in validation cohort," it might be clearer to say "mutational status is significantly associated with survival."
In the Conclusion section, the phrase "In this study, we identified three proteins" could be revised to "This study identified three proteins" for conciseness.
In the Conclusion, the phrase "In summary, through discovery proteomics in patient tissue" might be clearer as "In summary, this study employed discovery proteomics on patient tissue."
Apart from these minor suggestions, the text is well-written, and the grammar is correct.
Reviewer 2 Report
Comments and Suggestions for Authors
The research work entitled “CA9, CYFIP2 and LGALS3BP - a novel biomarker panel to aid prognostication in glioma” describes the potential role of new markers for glioma growth and progression investigations. The manuscript comprises an Introduction, materials, and method part along with characterization of selected markers and discussion. The work is novel and in the scope of the journal.
The comments are as follows:
In the introduction part, there is not enough information about the characterization of selected markers. The selected protein has some function in the pathogenesis of glioma, please clarify this and add that information to the Introduction.
In the materials and methods part, it will be beneficial for the reader to add subtitles to the manuscript.
The presentation of results, obtained from mass spectrometry will be clearer to present it as a heat map, and with proper statistics including PC.
Figure 3 presented a poor-quality of pic.
In the discussion part, there is still a lack of information about the colouration of selected authors' proteins and the survival rate of glioma patients. This should be added.
Reviewer 3 Report
Comments and Suggestions for Authors
Dear Editor,
this manuscript by Hudson et al identifies CA9, CYFIP2 and LGALS3BP proteins as potential novel prognostic biomarker for glioma disease.
Authors performed a proteomic screening starting from a discovery cohort of 6 IDH+ astrocytomas diagnosed as low-grade and recurred as high-grade. Among 32 proteins found to be differentially expressed in the matched samples analysis were restricted to five candidate biomarker according to literature reported function in glioma.
A validation cohort of 82 patients was then analyzed and main results can be summarized in the evidence that the panel of three (over the six ?) biomarker analyzed (i.e. CA9-CYFIP2-LGALS3BP) can be used as prognostic indicator in univariate analysis. However , in a multivariate analysis the individual proteins fail as significant prognostic biomarker.
Overall, the topic covered in this work is clearly interesting. Undoubtedly, the search for new biomarkers for the diagnosis and follow-up of such an aggressive and lethal disease as glioma is of utmost urgency. However, the conclusions reached by the authors appear slightly overestimated. The following points need to be addressed by the authors to strength their findings.
-it’s not clear whether candidate biomarker analyzed in the validation cohort are five or six
-authors should explain why in the discovery cohort CYFIP2 is downregulated while in the discovery cohort resulted to be associated to high grade gliomas
-CA9 data are missing in Table S4
-It’s mentioned Figure S3 (please change as Figure S2)
-It’s not clear how it’s calculated the prognostic value of the 3-protein panel. If the population is dichotomized into two groups, dividing patients with a level above or below the cut-off for the three biomarker panel, then the Kaplan Meyer would have a more important prognostic value.
-Although authors present an in silico analysis showing no interaction between the three biomarker, at least validation in a set of commercial available or patient derived cell lines would have strength the importance of these biomarker in GBM. In fact, the authors themselves discuss the importance of studying the possible interactions between these proteins in the future.
Round 2
Reviewer 3 Report
Comments and Suggestions for Authors
I would reccomend to publish the paper in the present form as the criticisms raised in the first round were addressed by the authors.